# Hardware-in-Loop Comparison of Physiological Closed-Loop Controllers for the Autonomous Management of Hypotension

**DOI:** 10.3390/bioengineering9090420

**Published:** 2022-08-27

**Authors:** Eric J. Snider, David Berard, Saul J. Vega, Evan Ross, Zechariah J. Knowlton, Guy Avital, Emily N. Boice

**Affiliations:** 1U.S. Army Institute of Surgical Research, JBSA Fort Sam Houston, San Antonio, TX 78234, USA; 2Trauma & Combat Medicine Branch, Surgeon General’s Headquarters, Israel Defense Forces, Ramat-Gan 52620, Israel; 3Division of Anesthesia, Intensive Care & Pain Management, Tel-Aviv Sourasky Medical Center, Tel-Aviv 64239, Israel

**Keywords:** control systems, hemorrhage shock, fluid resuscitation, closed loop, infusion, controllers, fluid management, hypotension, fuzzy logic, decision table

## Abstract

Trauma and hemorrhage are leading causes of death and disability worldwide in both civilian and military contexts. The delivery of life-saving goal-directed fluid resuscitation can be difficult to provide in resource-constrained settings, such as in forward military positions or mass-casualty scenarios. Automated solutions for fluid resuscitation could bridge resource gaps in these austere settings. While multiple physiological closed-loop controllers for the management of hypotension have been proposed, to date there is no consensus on controller design. Here, we compare the performance of four controller types—decision table, single-input fuzzy logic, dual-input fuzzy logic, and proportional–integral–derivative using a previously developed hardware-in-loop test platform where a range of hemorrhage scenarios can be programmed. Controllers were compared using traditional controller performance metrics, but conclusions were difficult to draw due to inconsistencies across the metrics. Instead, we propose three aggregate metrics that reflect the target intensity, stability, and resource efficiency of a controller, with the goal of selecting controllers for further development. These aggregate metrics identify a dual-input, fuzzy-logic-based controller as the preferred combination of intensity, stability, and resource efficiency within this use case. Based on these results, the aggressively tuned dual-input fuzzy logic controller should be considered a priority for further development.

## 1. Introduction

Traumatic injury is an important problem in both civilian and military contexts: it is a leading cause of death and disability worldwide, accounting for an estimated 8% of all deaths and approximately 10% of all disability adjusted life years in 2019 [1]. In particular, hemorrhage is estimated to account for nearly 40% of all trauma-related deaths, with the majority occurring in the first 24 h after injury [2]. In the recent American military experience, hemorrhage after traumatic injury is the leading cause of preventable death on the battlefield, accounting for 90% of all preventable combat fatalities [3].

The standard of care for the bleeding trauma patient includes rapid control of ongoing hemorrhage and fluid resuscitation to re-establish oxygen delivery to ischemic tissues by restoring cardiac output. Multiple endpoints to traumatic shock resuscitation have been described, including restoration of arterial blood pressure, but also clearance of accumulated lactate and restoration of normal hemostatic function [4]. Damage control resuscitation (DCR) refers to a specific resuscitation strategy used for treating bleeding casualties until complete control of the hemorrhage is achieved. DCR focuses on the restoration of hemostasis by emphasizing whole blood resuscitation (as opposed to crystalloids) and permissive hypotension (i.e., a systolic blood pressure target of 80–90 mmHg), balancing between the needs of providing sufficient tissue perfusion and avoiding exacerbation of the hemorrhage [5].

The delivery of goal-directed resuscitative care to the bleeding trauma patient is a cognitively demanding task, even for highly experienced providers [6]. Physiological closed-loop controllers (PCLCs) for resuscitation have been proposed as a potential solution to mitigate cognitive overload while improving the precision of care [7]; this benefit could be especially pronounced for situations in which care is highly specialized (such as burn resuscitation), or in mass-casualty scenarios when any given provider must care for multiple high-acuity patients [8,9]. In general, closed-loop control systems attempt to control a variable of interest around a setpoint by varying the intensity of an input. As an example, a thermostat controls the temperature of a room by varying the intensity of the air conditioning or heating. In the case of hemorrhagic shock resuscitation, the standard goal in clinical practice, as well as the one most suitable to serve as the setpoint for a PCLC, would be the patient’s arterial pressure, by modulating the intensity of fluid resuscitation; i.e., essentially, an automated version of goal-directed therapy which is the current recommended practice [10].

While multiple PCLCs for the resuscitation of hemorrhagic shock have been proposed [11], methods for comparing the performance of one controller to another remain limited. In this study we compared four controller types, each configured two ways, for a total of eight controller configurations. These four controller types, decision table [12,13], single-input fuzzy logic [14], dual-input fuzzy logic [15,16,17], and proportional–integral–derivative [18,19,20], have been previously used in PCLC applications but not compared head-to-head. The goal for this effort was to determine, as objectively as possible, which of the many possible controllers demonstrate the best performance, making it favorable for further development. Towards this, we use the HATRC test platform we previously developed for hardware-in-loop testing of controllers through a range of hemorrhage scenarios [21]. We propose a series of aggregate metrics tuned to different controller performance features for inter-controller comparisons across various testing scenarios.

## 2. Materials and Methods

### 2.1. Controller Design and Tuning

A total of eight controller types and configurations were evaluated using the HATRC test platform (see Section 2.2). Four controller types were used: decision table (DT), single-input fuzzy logic (SFL), dual-input fuzzy logic (DFL), and proportional–integral–derivative (PID) control. Two configurations of each controller type were obtained using a conservative and aggressive tuning methodology. Conservative tuning was guided by more rigorously avoiding overshooting the set point while allowing a larger rise time. Aggressive tuning prioritized minimizing the rise time above avoiding overshoot. Each controller was developed as follows.

#### 2.1.1. Decision Table

Decision Table logic was adapted from a prior study conducted by Marques et al. [22]. Logic was modified to a target pressure of 65 mmHg and each step in the logic was proportional to the max flow rate (Q_Max_) for HATRC infusion—500 mL/min—and target pressure (P_Target_). From previous work, the logic was found to be incapable of reaching the target in severe hemorrhage scenarios [21], so it was adjusted as highlighted in Table 1 for the aggressive and conservative DT tuning. Due to the nature of DT logic, it was not as objectively tuned as the other three controller types.

#### 2.1.2. Single-Input Fuzzy Logic

Fuzzy logic controllers were developed using the Fuzzy Logic Designer toolbox in MATLAB (MathWorks, Natick, MA, USA). All fuzzy logic controllers had a single output of infusion flow rate, and a Sugeno-type fuzzy inference system was used due to its improved computational efficiency and guaranteed continuity over the output surface compared to a Mamdani-type system [23]. For the SFL controllers, performance error (% of target pressure, with a value of 1 meaning the input is at the target) was used as the input, and the conservative/aggressive tuning was applied to the three input membership functions shown in Figure 1. For example, a system pressure of 32.5 mmHg with a set point of 65 mmHg would result in a performance error input value of 0.5. The membership functions of the conservative tuning would classify this as a “Low” pressure with a ~0.75 degree of membership while aggressive tuning would classify this as “VeryLow” with a 1 degree of membership. All outputs were linearly mapped to three values: Max (0–500 mL/min), Medium (0–250 mL/min), and Off (0 mL/min). The aggressive and conservative tuning configurations used the same three rules:(1)If *PerformanceError* is *VeryLow*, then *InfusionRate* is *Max*.(2)If *PerformanceError* is *Low*, then *InfusionRate* is *Med*.(3)If *PerformanceError* is *Set*, then *InfusionRate* is *Off*.

#### 2.1.3. Dual-Input Fuzzy Logic

For the two DFL controller configurations, performance error was again used as the first input to the system. The second input was the rate of the performance error change over the last three samples. This was chosen as the second input to account for cases where the pressure may be near the set point, resulting in a performance error ~ 1, but the presence of a large hemorrhage may prevent the controller from reaching the set point. The membership functions for these inputs are shown in Figure 2. Rules for the conservative tuning were as follows:(1)If *PerformanceError* is *VeryLow* or *(d/dt)PerformanceError* is *dropFast*, then *InfusionRate* is *Max*.(2)If *PerformanceError* is *Low* and *(d/dt)PerformanceError* is not *riseFast*, then *InfusionRate* is *Med*.(3)If *PerformanceError* is *Set* and *(d/dt)PerformanceError* is not *dropSlow*, then *InfusionRate* is *Off*.

The aggressive tuning utilized a modified set of rules:(1)If *PerformanceError* is *VeryLow* or *(d/dt)PerformanceError* is *dropFast*, then *InfusionRate* is *Max*.(2)If *PerformanceError* is *Low*, then *InfusionRate* is *Max*.(3)If *(d/dt)PerformanceError* is *dropSlow*, then *InfusionRate* is *Max*.(4)If *PerformanceError* is not *Set*, then *InfusionRate* is *Med*.(5)If *PerformanceError* is *Set*, then *InfusionRate* is *Off*.

#### 2.1.4. Proportional–Integral–Derivative Control

We previously developed PID controllers for automated hemorrhagic shock resuscitation using the HATRC platform [24]. Here, the same controllers were compared against additional controller types. Briefly, pressure to volume infusion relationships were established by analyzing hemorrhage data from a previous porcine animal study [25]. Utilizing the System Identification Toolbox in MATLAB, a plant model based on those pressure–volume relationships was obtained to tune an aggressive and conservative PID controller. As previously stated, conservative tuning was guided by more rigorously avoiding overshooting the set point while allowing a larger rise time, while aggressive tuning prioritized minimizing the rise time rather than avoiding overshoot.

### 2.2. HATRC Platform and Subject Variability

To physiologically imitate volume responsiveness of swine during various hemorrhaging scenarios, a Hardware-in-loop Automated Testbed for Resuscitation Controllers (HATRC) was utilized [21,24]. The HATRC consisted of a closed loop system, circulating water via a peristaltic pump at 145 mL/min and monitored through a pressure transducer (ICU Medical, San Clemente, CA, USA). The venous capacitance for the system was provided by two PhysioVessels (PVs) designed to hydrostatically emulate the empiric pressure–volume relationship seen in experimental swine hemorrhagic shock resuscitation data for both whole blood (PV_WB_) and crystalloid (PV_Crys_) infusates [26] These two PhysioVessels were integrated into the system via solenoid valves (Grainger, Lake Forest, IL, USA) and connected to two additional peristaltic pumps for fluid infusion and outflow. The PhysioVessels for these experiments were developed to avoid an unnecessarily complex physiological model but were restrictive to the underlying swine physiological data they modeled. These PhysioVessels were designed to simulate swine arterial pressure response to volume resuscitation in the pressure range between profound shock and the target pressure of DCR using either WB or Crystalloids; simulating anything outside of this range would require a different, more convoluted model. For subject variability, the PV_WB_ and PV_Crys_ can be substituted for additional PVs to alter the volume responsiveness of the system +/− 1 SD of the modeled swine data. The entire system, including infusion rates as determined by the controllers and dynamically calculated variable outflow rates, was fully controlled using MATLAB. Serial communication was used for pump control, and a USB interface (U3, LabJackCorp, Lakewood, CO, USA) was used for solenoid valve control. A data acquisition system (PowerLab, ADinstruments, Sydney, Australia) was used to capture “arterial” pressure waveform data at a rate of 40 Hz and MAP was calculated using a 5 s moving average.

An MAP of 65 mmHg was set as the target for controlled infusion for all scenarios, as this agreed with the Remote Damage Control Resuscitation (RDCR) guidelines [5]. Outflow from HATRC was a factor of urine and hemorrhage rates. Urine rate was held constant at 1.4 mL/min unless MAP fell below 50 mmHg, at which point urine output ceased [27]. Hemorrhage rate was a result of an MAP-dependent hemorrhage factor and a hemostasis factor. The hemorrhage factor was configured to be zero at 30 mmHg MAP and linearly increasing with MAP until reaching a maximum hemorrhage rate of 140 mL/min at an MAP of 65 mmHg. The hemostasis factor slowed the hemorrhage rate over time except for coagulopathy scenarios, in which case the hemostasis factor was zero. Lastly, to imitate the rupture of a clot, if MAP ever exceeded the target value by more than 5%, the hemostasis and hemorrhage factors would be reset to their initial values to increase the rebleed rate [25]. The hemostasis factor was determined through the analysis of the normalized swine hemorrhage data sets utilizing linear regression and R-squared values as described in a previous study [24].

### 2.3. Hemorrhage Scenarios for Testing

We previously developed 11 hemorrhage scenarios to challenge PCLCs using different infusates, starting MAP values, and differing hemorrhage rates simulating trauma relevant injuries [24]. For testing of the eight PCLC types and configurations in this new study, that initial list of 11 scenarios was down selected to 4. These scenarios focused exclusively on resuscitation with WB, as this is the preferred infusate for DCR [5] and prior PID controller evaluation showed similar performance for WB vs. Crystalloid [24]. Scenario 1 was the only 62 min-long scenario, designed to simulate a severe, compressible hemorrhage that has been successfully controlled prior to beginning resuscitation (e.g., tourniquet application on a limb hemorrhage). Therefore, this scenario had a low starting MAP with no active bleed. The controller was allowed to attempt to reach the target MAP for 30 min. At the 30 min mark, a severe bleed was triggered to imitate hemorrhage recurrence (e.g., slippage of a tourniquet) that was subsequently corrected after 2 min, and the system was allowed to equilibrate again during the remaining 30 min of the scenario. All other scenarios were 30 min long. Scenarios 2 and 3 were designed to simulate a non-compressible hemorrhage, only slowed by internal hemostatic mechanisms. The difference between these two scenarios was the initial MAP: Scenario 2 began at 65 mmHg to mimic a resuscitation starting at a higher initial point (compensated shock), and Scenario 3 began at 45 mmHg simulating profound hypotension. Both of these scenarios simulated an ongoing bleed with a severe initial hemorrhage rate and included hemostasis in the simulation to slow the bleeding rate over time. Lastly, Scenario 4 incorporated coagulopathy (caused, for instance, by hypothermia [28]) for increased complexity, manifested as gradual failure of internal hemostatic mechanisms after 5 min, allowing continued, exacerbating massive hemorrhaging. Scenario 4 began at 45 mmHg with an initial bleed that slowed due to hemostasis factors until the 5 min mark. Hemostasis was then gradually reduced, and the hemorrhaging rate increased to a maximum rate (140 mL/min) until the end of the scenario.

### 2.4. Performance Evaluation

A series of metrics was used to assess the performance of each PCLC in handling the four test scenarios. Some of these were as described by Varvel [29], Mirinejad [30], Marques [22], and the International Electrotechnical Commission [31], others are modifications of those same metrics, and another set we introduced in a previous work [21,24]. For aggregate metrics, unitless or percentage-based metrics were preferred, so normalization methods are detailed below when they were required. A summary of these metrics is as follows:Median Performance Error (MDPE): the median of the measured errors relative to target pressure (%, lower is better).Median Absolute Performance Error (MDAPE): the median of the absolute value of the measured errors (%, lower is better).Target Overshoot: the maximum pressure measured relative to the target (%, lower is better).Effectiveness: the percent of time that the pressure stayed within ±5 mmHg of the target pressure (%, higher is better).Wobble: the median of the absolute difference for each measured error and MDPE (%, lower is better)End-state Divergence: the slope of the linear fit of measured error vs. time during the final 10% segment of the test scenario (%/hr). For simplicity, the absolute value of the End-state Divergence was used, and the units were removed by multiplying the total length (in hours) of each scenario (%, lower is better).Percent Rise Time: the time required for measured pressure to reach 90% of the target value (min). This metric was made proportional to the total scenario time, in minutes (%, lower is better).Volume Efficiency: the ratio of the total input volume to the total output volume during the test scenario (no units, lower is better)Areas Above and Below Target: in a plot of pressure vs. time, these are the sum of the areas between the measured pressure and the target pressure lines, normalized by the target pressure and kept separated as those that lie above and below the target line, respectively (min). These metrics were further normalized to the total scenario time to make them proportional to the highest possible area above or below the target (%, lower is better).Mean Infusion Rate: the mean rate of infusion across the scenario window relative to the maximum (500 mL/min) possible infusion rate (%, lower is better)Infusion Rate Variability: the standard deviations of infusion rates calculated in 2 min-long segments, first averaged and then normalized using the mean infusion rate (%, lower is better)

In addition to these performance metrics, we introduce here an additional one termed ‘MDAPE at Steady State’ (MDAPE_SS_). This is simply the MDAPE as described above, calculated after the system has reached steady state. This modified measurement attempts to describe the longer-term accuracy of the controller by looking past the scenarios’ initial conditions after the system has stabilized.

### 2.5. Aggregate Metric Methodology

The described testing method created an abundance of data and performance metrics that can be overwhelming, impeding conclusion formation and decision making for controller design. For the purpose of making the results more accessible and concise, performance metrics representing general attributes of the controllers were collated into three groups, titled:Intensity—how good the controller is at rapidly and effectively treating hypotension. This group included effectiveness, area-below-target, and percent rise time as shown in Equation (1).
(1)$$Intensity=\frac{PercentRiseTime\times AreaBelowTarget}{Effectiveness}$$

Stability—how the controller performs in maintaining a stable state and minimizing overshoot. This group included MDAPE_SS_, wobble, target overshoot, area-above-target, and absolute value of end-state divergence as shown in Equation (2). Preliminary analysis revealed that this metric was not sufficiently penalized for not reaching target pressure, so MDAPE_SS_ was squared in the aggregate calculation to address this, as it is the only individual metric in the aggregate that reflects this impact.


(2)
$$Stability=Wobble\times \left|End\text{-}stateDivergence\right|\times \left(AreaAboveTarget+TgtOvershoot\right)\times {\left(MDAPE_{SS}\right)}^{2}$$


Resource Efficiency—the controller’s ability to minimize consumption of fluid and hardware wear-and-tear. This group included infusion rate variability, mean infusion rate, and volume efficiency as shown in Equation (3).


(3)
Resource Efficiency=MeanInfusionRate×InfusionRateVariability×VolumeEfficiency


For each metric, a low score means better controller performance. Aggregate scores were only calculated for average performances across the four scenarios. To normalize the weights of each underlying metric, each performance metric was normalized to its median across the 8 PCLC types and configurations prior to aggregate metric calculation. Each controller received a score in all three aggregate metrics. An average score was obtained by calculating an average of the intensity, stability, and resource efficiency aggregate metrics, where each aggregate was weighed evenly. A significant limitation in this approach is the lack of clinical knowledge to determine the physiologic and clinical implications of these metrics, namely “what is the optimal resuscitation profile?” Further clinical and pre-clinical research is required to answer this question, which is key to the design and evaluation of any PCLC fluid resuscitation system.

Lastly, we investigated correlations between the aggregate metrics—intensity, stability, and resource efficiency—as well as the average aggregate score vs. each individual performance metric. This was performed using Prism 9.3.1 (GraphPad, San Diego, CA, USA) by a performing linear regression between each pair of aggregate metric and corresponding individual metric for each controller type and configuration (*n* = 8). Coefficients of determinations (R^2^) were compiled in a heat map and a correlation matrix was used to evaluate which individual metric correlated to aggregate scores.

## 3. Results

### 3.1. Scenario 1 Controller Performance

We first evaluated each controller using a scenario where the MAP was 45 mmHg and no active bleed was present, such as a compressible hemorrhage that had been controlled. After 30 min, a severe re-bleed occurred for 2 min followed by no ongoing hemorrhage—mimicking re-achievement of hemorrhage control. Representative results for the aggressive and conservative PID controller are shown for MAP and inflow/outflow vs. time (Figure 3A,B). Similar results for the other controllers can be found in Appendix A. Without an active bleed, all controllers were able to reach the target effectively, but the aggressive configurations were typically overshooting the target. Looking at the percent rise time, the aggressive configurations of all the controllers outperformed their conservative configuration as anticipated (Figure 3C). Comparing across controller types, the aggressive DFL and DT reduced percent rise time the most (aggressive DT 1.81% vs. SFL 4.07% vs. DFL 2.54% vs. PID 3.38%). The conservative SFL had the slowest rise time of all controller types tested at 7.87%. To quantify the overshoot effect, the area above the target metric highlighted the larger results for the aggressive configurations compared to the conservative (Figure 3D). All controllers overshot the target except the SFL conservative. Scenario 1 results for all performance metrics for each controller are shown in Appendix A.

### 3.2. Scenario 2 Controller Performance

Next, Scenario 2 involved an active hemorrhage that reduced its rate with time and an initial MAP of 65 mmHg—simulating a re-bleed event after initial stabilization. MAP and inflow/outflow vs. time plots are shown for the aggressive and conservative configurations of the SFL controller (Figure 4A,B). Plots for the remaining controller types and configurations are provided in Appendix A. With an active hemorrhage, the scenario resulted in more dynamic pump rates, with infusion rate variability being as high as 167% for the aggressive DT (Figure 4C). Interestingly, the conservative DT also had higher infusion rate variability than other controller types, indicating that the DT logic had inherently high scores for this metric, within this scenario. Each controller responded quickly to prevent large drops in MAP for this scenario, but the aggressive configuration continued to do so more quickly (Appendix A). As a result, the area below the target pressure was minor for all controllers, with both DT configurations and DFL aggressive having the smallest metric values (Figure 4D). The Scenario 2 results for all performance metrics for each controller type and configuration are shown in Appendix A.

### 3.3. Scenario 3 Controller Performance

Scenario 3 was similar to the previous scenario except the starting MAP was set to 45 mmHg, simulating acute resuscitation with an active, non-compressible hemorrhage. Representative results are shown for aggressive and conservative configured DT controller types, for MAP and inflow/outflow rate vs. time (Figure 5A,B). Plots for the other controller types and configurations can be found for Scenario 3 in Appendix A. This scenario presented wider discrepancies between controller configurations than the first two scenarios as a result of the initial distance from the target MAP and active hemorrhage slowing the recovery rate. This was best reflected by the area below the target pressure metric where both DT configurations, DFL aggressive, and PID aggressive outperformed the other controller configurations and types (Figure 5C). Another metric to highlight for this scenario was MDAPE at steady state which measured the error the controllers settled at (Figure 5D). The SFL conservative controller had the highest error at 9.5%, indicating that it was far from the target pressure at steady state. The SFL conservative controller had an average effectiveness of less than 8% for the 30 min scenario window, while all other controllers reached ±5 mmHg of the target for more than 75% of the scenario window (Appendix A). Scenario 3 results for all performance metrics for each controller type and configuration are shown in Appendix A.

### 3.4. Scenario 4 Controller Performance

The final scenario mimicked Scenario 3 except that active clotting ceased after 5 min, allowing coagulopathy to set in, which gradually increased the hemorrhage rate to the maximum, severe level and held at that rate for the remainder of the scenario. This was the most challenging to resuscitate against as there was a severe hemorrhage present the entire 30 min. This exaggerated scenario, although unlikely to be encountered in a clinical setting, was intentionally chosen as more of a stress-test to evaluate the limits of the controllers. Representative results are shown for both DFL controller configurations for MAP and inflow/outflow vs. time (Figure 6A,B). Results for all other controller configurations and types can be found in Appendix A. Most controllers reached steady state at different distances from the target MAP as can be seen in the DFL controllers MAP vs. time plots. This was reflected by the MDAPE at steady state with the aggressive configurations of the DT, DFL, and PID controllers outperforming the others (Figure 6C). This result was further shown with the effectiveness metric, as both SFLs, DFL conservative, and PID conservative had values below 20% while the rest of the controllers remained above 70% (Figure 6D). Scenario 4 results for all performance metrics for each controller type and configuration are shown in Appendix A.

### 3.5. Overall Controller Performance

#### 3.5.1. Performance Metrics

We evaluated overall, average controller performance across all scenarios. Average results for all performance metrics for each controller type and configuration are shown in Appendix A. For percent rise time, both DTs and DFL aggressive performed the best, with the SFL conservative having the worst performance (Figure 7A). Infusion rate variability highlighted how drastically flow rates were modified through the scenario and the reverse data trend was mostly observed, with both DTs altering pump settings most substantially, followed by the PID aggressive (Figure 7B). However, the data trend was not reversed for the DFL aggressive which had a low infusion rate variability. End-state divergence highlighted how stable the controller was at each scenario’s end. The DFL aggressive and both PIDs had the highest performance for this metric (Figure 7C). Lastly, effectiveness highlighted how much of each scenario’s duration the controller was within 5 mmHg of the target. As the SFL conservative consistently struggled to reach the target through the later scenarios, it had the worst performance with respect to this metric (Figure 7D). The aggressive configurations in general retained higher overall effectiveness as they more rapidly reached the target MAP.

#### 3.5.2. Aggregate Metrics Results

As is evident when examining the individual performance metrics, performance for each controller varied depending on which metric was selected for analysis. In order to make the controller comparison more accessible and support decision making, we created three aggregate metrics composed of a number of individual metrics that focused on certain controller performance features. Of note, a low score on each metric is preferred and indicates that the controller excelled for that specific measurement. Descriptions of each aggregate metric are detailed in Section 2.5. The results for the intensity aggregate are shown in Figure 8A. The best performing controllers for this aggregate were both DTs, DFL aggressive, and PID aggressive; all had values below one. Results for the stability aggregate are shown in Figure 8B. The best performers for this aggregate were the DT aggressive and PID conservative. Results for the resource efficiency aggregate are shown in Figure 8C. The best performers for this aggregate were both SFLs and DFL conservative.

Across all aggregate metrics, the DFL aggressive performed most consistently (Figure 9A) and had the lowest average score across all three aggregates (Figure 9B). Lastly, each aggregate metric and the overall average scores were evaluated for correlation to individual metrics. This was completed to identify if a single performance metric was sufficient for predicting the aggregate metric score, superseding the need for new aggregate scoring metrics (Figure 9C). The intensity score correlated strongly to a range of individual metrics including some that were not even included in the aggregate calculation (MDAPE at steady state). Three metrics had R^2^ correlations to the intensity greater than 0.80: MDAPE at steady state, effectiveness, and percent rise time. The stability metric had no strong correlation to any individual metric. The strongest correlation was to wobble at a 0.58 R^2^ value. Resource efficiency correlated strongly to infusion rate variability only (R^2^ = 0.997). The average of the aggregates had R^2^ values above 0.80 for three metrics: MDAPE at steady state, effectiveness, and percent rise time, the same metrics as the intensity aggregate.

## 4. Discussion

Resuscitation of a patient in hemorrhagic shock is a very demanding task, requiring constant attention and patient monitoring. This level of concentration might not be readily available in resource austere conditions, such as in mass casualty incidents or in forward military positions. If properly constructed, medical devices such as PCLCs for managing hypotension, hold the potential to greatly improve resuscitation outcomes. While multiple PCLCs have been investigated, how to assess controller performance and thus choose optimum PCLCs for advanced development remains unclear.

In this work, we developed and configured several controller types for evaluation in our hardware-in-loop testing platform for simulated hemorrhagic shock scenarios. Assessment of these controllers included traditional controller-type performance metrics, as well as newly developed indicators focused on how well the simulated patient was resuscitated. The combination of this physical testbed, simulated trauma scenarios, and key performance indicators provides a streamlined platform for comparison of a large number of controller types and tunings for identification of systems for further animal and clinical testing.

Evaluating controllers using 13 single performance metrics identifies individual controllers that perform both well and sub-optimally in each scenario. This discrete analysis limits an overall decision about optimal controller choice, as some controllers perform well in one scenario and badly in others. This pattern of irresolution prompted the development of aggregate performance metrics. These new metrics combine subsets of the originals to allow for examination of multiple dimensions of the controller performance. These aggregate metrics can therefore be centered around how well the controller resuscitates the simulated patient (intensity), how well the controller monitors and maintains the simulated patient at setpoint (stability), and how well the medical device performs (resource efficiency).

By allowing for a higher-level, more comprehensive assessment of controller performance, these aggregate metrics permitted us to identify the aggressive configuration of the DFL controller for advanced development. This controller type and configuration performed well in all three aggregate metrics, when averaged across all scenarios.

The different scenarios presented here were designed to represent diverse real-world clinical circumstances that a PCLC system might encounter. As might be expected, the PCLC systems demonstrated variable performance across different scenarios. These differences in PCLC performance suggests that specific PCLCs are better suited to specific clinical circumstances compared to others. However, the optimal resuscitation profile for these different hemorrhage scenarios is not known. Further, the underlying reasons for the differences in performance are not immediately clear from this data either. Future progress in this line of research needs to address what the optimal resuscitation profile is. Answers to this question along with developing a deeper understanding of controller performance characteristics will allow for a better alignment of PCLC type with patient conditions such that the optimal controller is engaged for the patient’s present condition.

The results presented here should be interpreted in view of certain limitations. First, while our HATRC benchtop circulatory system is modeled on empirical physiological data, like any in vitro or in silico solution, it is unable to replicate exactly the physiology of a living organism—planned future work will address this limitation by deploying selected PCLCs in an in vivo model of hemorrhage and resuscitation. Second, our aggregate metrics were formulated based on an application-oriented, rather than a mathematical, approach; alternative aggregate metrics certainly exist and may come to different conclusions. While doing so is beyond the scope of the present effort, future work may focus on developing aggregate performance metrics based on mathematical analysis of controller performance. Third, controllers were tuned as aggressive or conservative against similar goals, but this process is subjective and there are an infinite number of controller tunings that can be evaluated. The downselected DFL controller will be further tuned using HATRC prior to in vivo deployment. Finally, while this paper presents the functional characteristics of multiple PCLCs across different clinically relevant hemorrhage scenarios, it was beyond the scope of this effort to directly compare the performance of the PCLC against the performance of a human operator; accordingly, we cannot make any inferences as to the relative effectiveness of the PCLC systems compared to the current standard of care. Future iterations of this project may involve direct comparison of human operators against a PCLC system in clinically and militarily relevant simulated patient management scenarios.

## 5. Conclusions

From these data, we conclude that it is possible to compare PCLC performance across multiple simulated hemorrhage conditions. However, results are hard to track across the wide range of performance metrics. Instead, we identified aggregated metrics to simplify data interpretation and identified an aggressively configured DFL controller as the best combination of intensity, stability, and resource efficiency, and to therefore recommend it for future development.

## Figures and Tables

**Figure 1 bioengineering-09-00420-f001:**
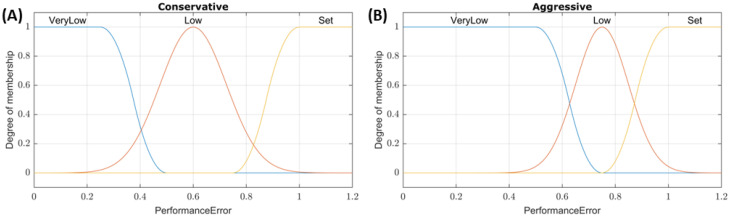
Membership function plots for SFL controllers. Plots of the *PerformanceError* input membership functions for (**A**) conservative and (**B**) aggressive tuned single-input fuzzy logic controllers.

**Figure 2 bioengineering-09-00420-f002:**
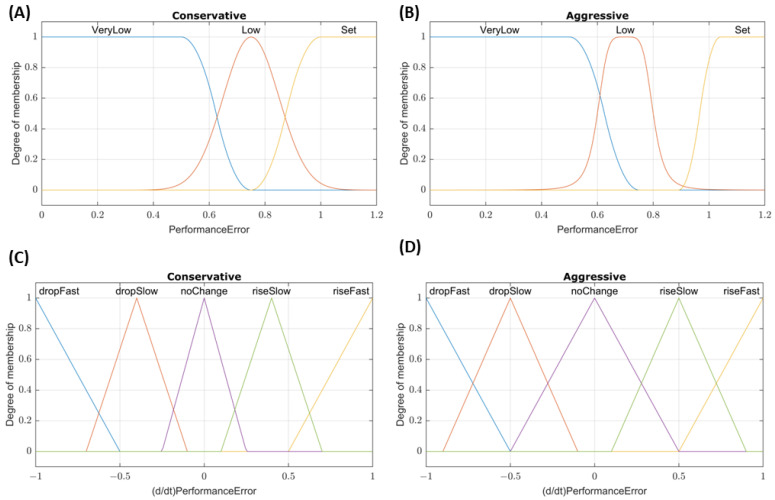
Membership function plots for DFL controllers. Plots of the *PerformanceError* and *(d/dt) PerformanceError* input membership functions for (**A**,**C**) conservative and (**B**,**D**) aggressive tuned dual-input fuzzy logic controllers, respectively.

**Figure 3 bioengineering-09-00420-f003:**
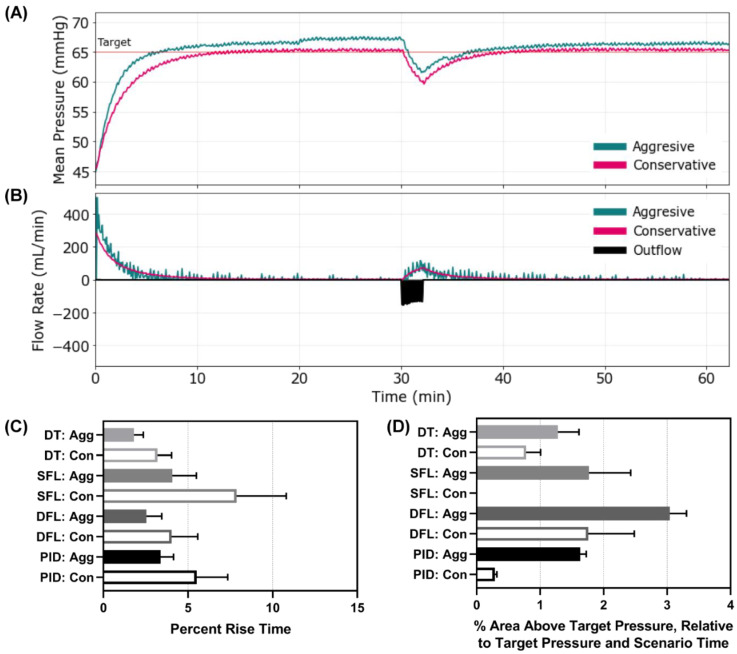
Controller performance for Scenario 1. Representative (**A**) MAP and (**B**) inflow/outflow vs. time for a single run of the aggressive and conservative PID controllers through Scenario 1. Outflow rates are only shown for one controller run. In Scenario 1, MAP begins at 45 mmHg with no active hemorrhage until at 30 min when a severe hemorrhage occurs for 2 min, followed by a return to no hemorrhage. Performance results for each controller type (DT = decision table; SFL = single-input fuzzy logic; DFL = dual-input fuzzy logic; PID = proportional–integral–derivative controller) and configuration (Agg = aggressive; Con = conservative) for the (**C**) percent rise time (lower is better) and (**D**) area above the target pressure metric (lower is better). Average results are shown for three subject variability runs for each. Error bars denote the standard deviation.

**Figure 4 bioengineering-09-00420-f004:**
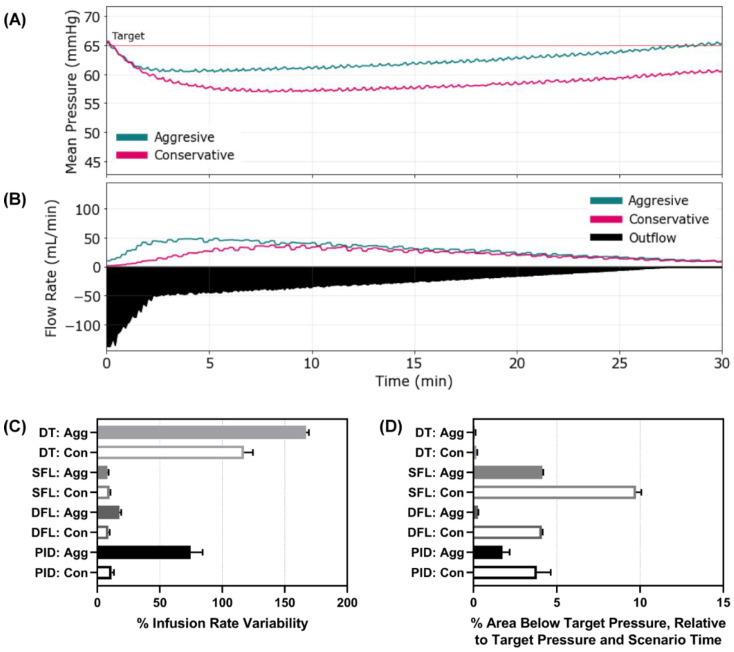
Controller performance for Scenario 2. Representative (**A**) MAP and (**B**) inflow/outflow vs. time for a single run of the aggressive and conservative single-input fuzzy logic controller runs through Scenario 2. Outflow rates are only shown for one controller run. In Scenario 2, MAP begins at a stable 65 mmHg and presents a severe hemorrhage that clots over the 30 min test scenario. Performance results for each controller type (DT = decision table; SFL = single-input fuzzy logic; DFL = dual-input fuzzy logic; PID = proportional–integral–derivative controller) and configuration (Agg = aggressive; Con = conservative) for the (**C**) percent infusion rate variability (lower is better) and (**D**) area below the target pressure metric (lower is better). Average results are shown for three subject variability runs for each. Error bars denote the standard deviation.

**Figure 5 bioengineering-09-00420-f005:**
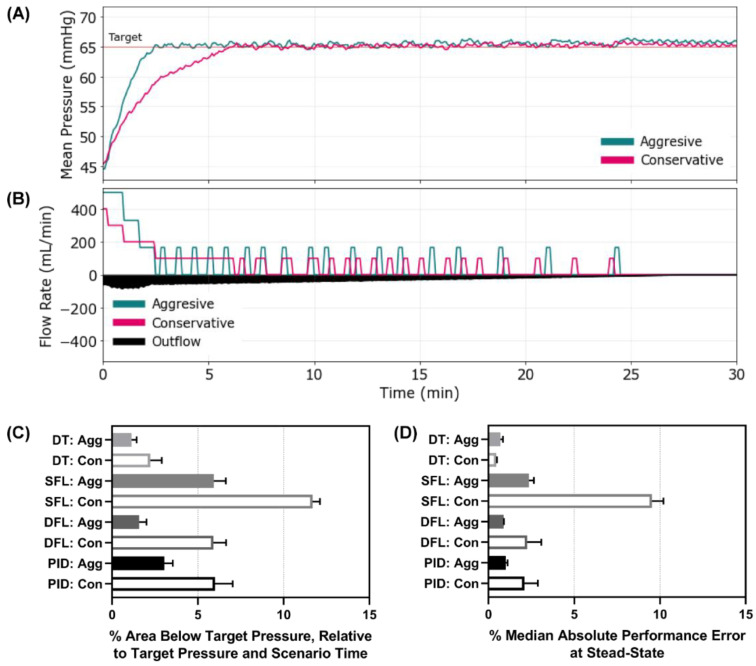
Controller performance for Scenario 3. Representative (**A**) MAP and (**B**) inflow/outflow vs. time for a single run of the aggressive and conservative decision table controllers through Scenario 3. Outflow rates are only shown for one controller run. In Scenario 3, MAP begins at 45 mmHg with an active hemorrhage clotting over the 30 min test run. Performance results for each controller type (DT = decision table; SFL = single-input fuzzy logic; DFL = dual-input fuzzy logic; PID = proportional–integral–derivative controller) and configuration (Agg = aggressive; Con = conservative) for the (**C**) area below target pressure (lower is better) and (**D**) median absolute performance error (MDAPE) at steady state (lower is better). Average results are shown for three subject variability runs for each. Error bars denote the standard deviation.

**Figure 6 bioengineering-09-00420-f006:**
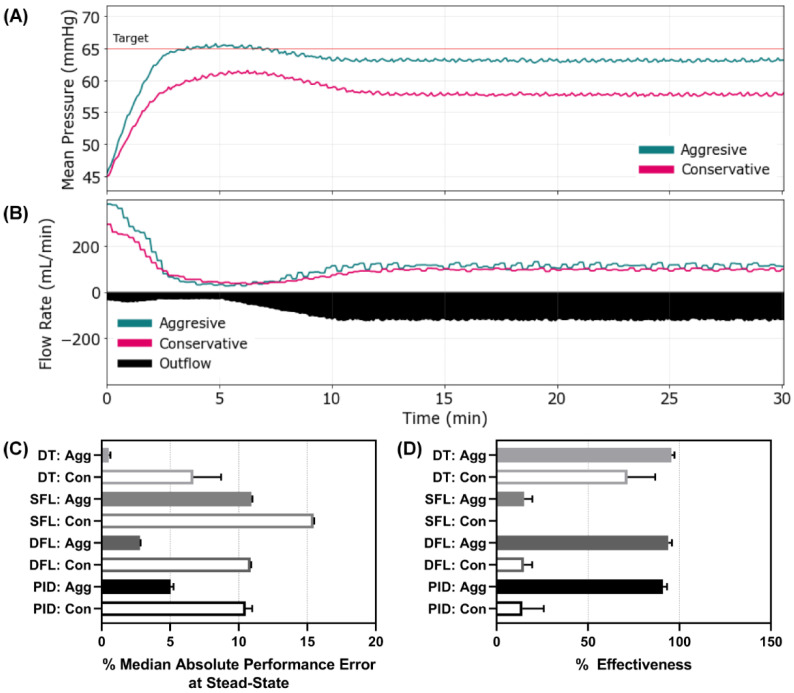
Controller performance for Scenario 4. Representative (**A**) MAP and (**B**) inflow/outflow vs. time for a single run of the aggressive and conservative dual-input fuzzy logic controller run through Scenario 4. Outflow rates are only shown for one controller run. In Scenario 4, MAP begins at 45 mmHg with an active hemorrhage that initially clots until the 5 min timepoint where clotting mechanisms were halted, and hemorrhage rates increased. Performance results for each controller type (DT = decision table; SFL = single-input fuzzy logic; DFL = dual-input fuzzy logic; PID = proportional–integral–derivative controller) and configuration (Agg = aggressive; Con = conservative) for the (**C**) median absolute performance error (MDAPE) at steady state (lower is better) and (**D**) effectiveness (higher is better). Average results are shown for three subject variability runs for each. Error bars denote the standard deviation.

**Figure 7 bioengineering-09-00420-f007:**
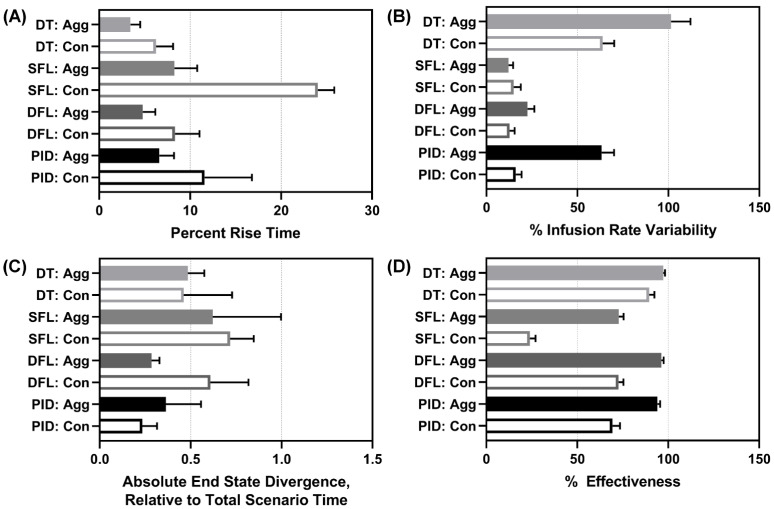
Average performance metrics results. Mean performance results for each controller type (DT = decision table; SFL = single-input fuzzy logic; DFL = dual-input fuzzy logic; PID = proportional–integral–derivative controller) and configuration (Agg = aggressive; Con = conservative) for (**A**) percent rise time (lower is better), (**B**) infusion rate variability (lower is better), (**C**) end-state divergence (lower is better), and (**D**) effectiveness (higher is better). Average results are shown for three subject variability runs. Error bars denote the standard deviation.

**Figure 8 bioengineering-09-00420-f008:**
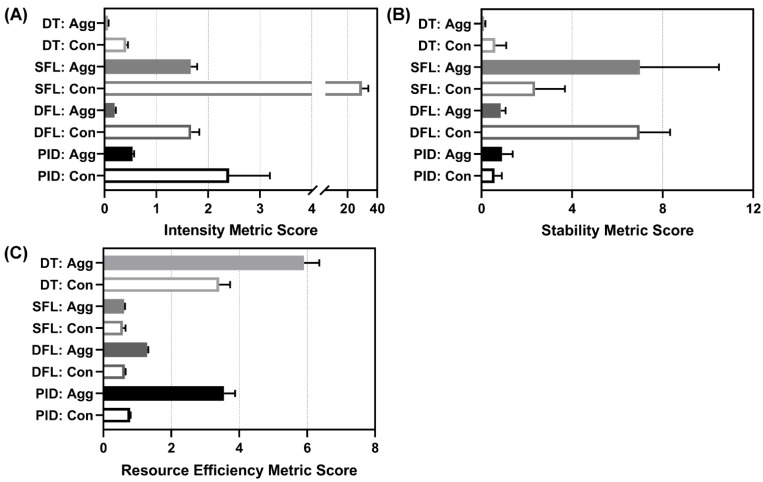
Aggregate metrics for each controller. Results for three developed aggregate metrics are shown for each controller type (DT = decision table; SFL = single-input fuzzy logic; DFL = dual-input fuzzy logic; PID = proportional–integral–derivative controller) and configuration (Agg = aggressive; Con = conservative): (**A**) intensity (note the broken *x*-axis), (**B**) stability, and (**C**) resource efficiency (lower is better for each). Each metric component is detailed in Section 2.5. Average results are shown for three subject variability runs. Error bars denote the standard deviation.

**Figure 9 bioengineering-09-00420-f009:**
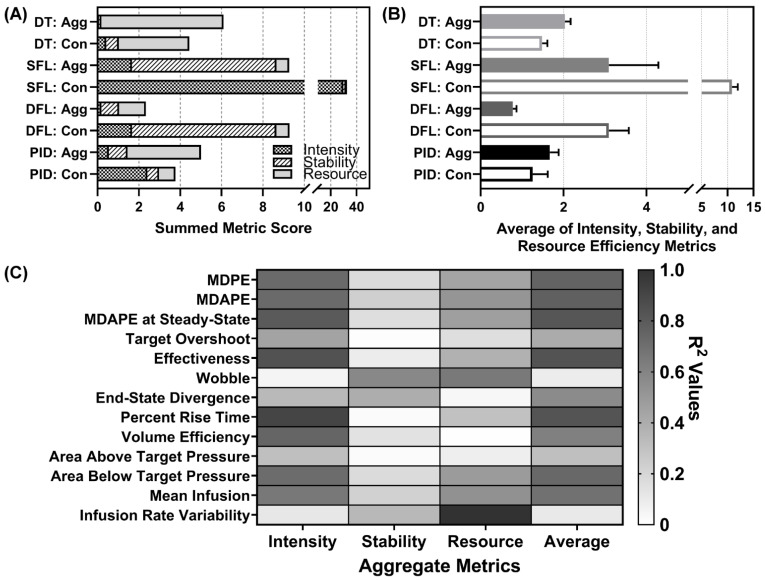
Average metric scores and correlation. (**A**) Stacked bar graph for all three aggregate metrics to highlight the various scores for each controller type (DT = decision table; SFL = single-input fuzzy logic; DFL = dual-input fuzzy logic; PID = proportional–integral–derivative controller) and configuration (Agg = aggressive; Con = conservative) (lower is better). (**B**) Average metric of intensity, stability, and resource efficiency metrics for each controller type and configuration (lower is better). Note the broken *x*-axis for each plot. (**C**) Correlation matrix for each individual performance metric vs. intensity, stability, and resource efficiency metrics. In addition, correlations were performed against the average aggregate metric score. Heat map colors reflect R^2^ values for each linear fit correlation, with darker grey indicating a stronger correlation.

**Table 1 bioengineering-09-00420-t001:** Overview of decision table controller logic for aggressive and conservative tuning. P_Target_ is the target pressure, which was set at 65 mmHg throughout, while P is the current pressure. Q_Max_ is the maximum possible infusion rate of the test platform which was set at 500 mL/min throughout.

Aggressive Decision Table	Conservative Decision Table
If P ≤ 85% P_Target_, Q = Q_Max_	If P ≤ 60% P_Target_, Q = Q_Max_
If P ≤ 95% P_Target_, Q = 66% Q_Max_	If P ≤ 70% P_Target_, Q = 80% Q_Max_
If P ≤ P_Target_, Q = 33% Q_Max_	If P ≤ 80% P_Target_, Q = 60% Q_Max_
If P > P_Target_, Q = 0	If P ≤ 90% P_Target_, Q = 40% Q_Max_
	If P ≤ P_Target_, Q = 20% Q_Max_
	If P > P_Target_, Q = 0

## Data Availability

The datasets generated during and/or analyzed during the current study are available from the corresponding author upon reasonable request.

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
