# Peer review of "Hardware-in-Loop Comparison of Physiological Closed-Loop Controllers for the Autonomous Management of Hypotension"

_bioengineering, 2022, doi:10.3390/bioengineering9090420_

Round 1

Reviewer 1 Report

This study compared the performance of different types of existing closed-loop systems for fluid resuscitation. It gave a description of the performance of different closed-loop controllers on a simulated system using several different scenarios mimicking hemorrhage. As the metrics data showed, some controllers performed well for one metric but not well for other metrics. Although this work is understood as an application study of the simulators, I wonder if the authors can at least discuss the reasons behind these disparities given the different switch-on conditions of different controllers to provide some new insights. If these scenarios were to relate to clinical treatment decisions, could the authors give some recommendations regarding which type of controller or choice of scheme is preferred for different conditions of hemorrhage. Furthermore, if one purpose of this paper is to highlight the functionality of the closed-loop system, did the authors use the test platform to compare the traditional fluid resuscitation systems and the closed-loop system.  How was the improvement of the closed-loop system?

Reviewer 2 Report

Hardware-in-Loop Comparison of Physiological Closed-Loop Controllers for the Autonomous Management of Hypotension

This paper conducted a comparative evaluation of closed-loop fluid resuscitation controllers.  The paper is written well overall, and deals with an important topic in biomedical engineering: medical autonomy.

I have a few minor comments for the authors to consider:

1. The paper considered non-model-based controllers only.  I would like the authors to consider discussing why model-based control options were not considered in the current work, such as adaptive control:

M. Alsalti, A. Tivay, X. Jin, G.C. Kramer, J.O. Hahn, “Design and In Silico Evaluation of a Closed-Loop Hemorrhage Resuscitation Algorithm with Blood Pressure as Controlled Variable,” ASME Journal of Dynamic Systems, Measurement and Control, Vol.144, No. 2, Article Number 021001, February 2022.

X. Jin, R. Bighamian, J.O. Hahn, “Development and In Silico Evaluation of a Model-Based Closed-Loop Fluid Resuscitation Algorithm,” IEEE Transactions on Biomedical Engineering, Vol. 66, No. 7, pp. 1905-1914, July 2019.

2. Lines 131-145: I suggest the authors show all the fuzzy IF-THEN rules for all the possible input combinations.  Specifically, there are 3 input 1 possibilities and 5 input 2 possibilities, so I feel that there should be 15 rules in total if I am not mistaken.  For example, what is the response of the DFL control to “VERY LOW” and “RISE FAST”?
